# Competing charge-density wave instabilities in the kagome metal ScV$_6$Sn$_6$

Saizheng Cao[1], Chenchao Xu [2], Hiroshi Fukui [3], Taishun Manjo[3], Ying Dong [4], Ming Shi [1,5], Yang Liu [1], Chao Cao [1] ✉ & Yu Song [1] ✉

Owing to its unique geometry, the kagome lattice hosts various many-body quantum states including frustrated magnetism, superconductivity, and charge-density waves (CDWs). In this work, using inelastic X-ray scattering, we discover a dynamic short-range $\sqrt{3} \times \sqrt{3} \times 2$ CDW that is dominant in the kagome metal ScV$_6$Sn$_6$ above $T_{CDW} \approx 91$ K, competing with the $\sqrt{3} \times \sqrt{3} \times 3$ CDW that orders below $T_{CDW}$. The competing CDW instabilities lead to an unusual CDW formation process, with the most pronounced phonon softening and the static CDW occurring at different wavevectors. First-principles calculations indicate that the $\sqrt{3} \times \sqrt{3} \times 2$ CDW is energetically favored, while a wavevector-dependent electron-phonon coupling (EPC) promotes the $\sqrt{3} \times \sqrt{3} \times 3$ CDW as the ground state, and leads to enhanced electron scattering above $T_{CDW}$. These findings underscore EPC-driven correlated many-body physics in ScV$_6$Sn$_6$ and motivate studies of emergent quantum phases in the strong EPC regime.

Quantum materials are typically strongly correlated or topologically nontrivial, giving rise to unconventional superconductivity[1–5], electronic nematicity[6,7], topological phases of matter[8,9], and quantum criticality[10,11]. A common hallmark of quantum materials is the presence of competing electronic instabilities, such as the competition between a ferromagnetic metal and a paramagnetic insulator in the manganites that lead to colossal magnetoresistance[12,13], and the competition between charge-density wave (CDW) and superconductivity in the cuprates[14–17].

Whereas the physics in many quantum materials are derived from strong electronic correlations, the unique geometry of the kagome lattice leads to geometric frustration, Dirac cones, magnon/electronic topological flat bands, and van Hove singularities[18–27], the combination of which gives rise to nontrivial electronic topology and correlated many-body states. As exemplified by $A$V$_3$Sb$_5$ ($A$ = K, Rb, Cs)[28] and FeGe[29,30], kagome metals could exhibit an unconventional CDW breaking both time-reversal and rotational symmetries[31–35] coexistent with a superconducting ground state[31,36,37], and a CDW that coexists with antiferromagnetism that enhances the ordered moment[29,30],

demonstrating the kagome lattice to be amenable to unconventional CDWs. Furthermore, the CDWs in both $A$V$_3$Sb$_5$ and FeGe are associated with a $2 \times 2$ in-plane ordering[31,38–41], indicating a prominent role of nesting between neighboring van Hove singularities[42–44].

Recently, CDW was discovered in the bilayer kagome metal ScV$_6$Sn$_6$[45], a member of the HfFe$_6$Ge$_6$-type compounds. Similar to $A$V$_3$Sb$_5$, V atoms in ScV$_6$Sn$_6$ form kagome layers with V-V distances in the range 2.73–2.75 Å, the V $d$-orbital bands cross the Fermi level, and there are no local moments[45]. In contrast to $A$V$_3$Sb$_5$ and FeGe, the V atoms in ScV$_6$Sn$_6$ form kagome bilayers [Fig. 1a], and the CDW is associated with a $\sqrt{3} \times \sqrt{3}$ in-plane ordering [Fig. 1b], and a tripling of the unit cell along the $c$-axis. Furthermore, whereas the CDW in $A$V$_3$Sb$_5$ is dominated by in-plane displacements of the V atoms[40] and hosts a superconducting ground state, the CDW in ScV$_6$Sn$_6$ is mostly driven by displacements of the Sc and Sn atoms along the $c$-axis[45], and no superconductivity is observed up to pressures of 11 GPa[46]. Optical reflectivity measurements and electronic structure calculations indicate that the CDW in ScV$_6$Sn$_6$ is unlikely to result from Fermi-surface nesting, and the CDW does not exhibit a prominent

[1]Center for Correlated Matter and School of Physics, Zhejiang University, 310058 Hangzhou, China. [2]School of Physics, Hangzhou Normal University, 310036 Hangzhou, China. [3]Japan Synchrotron Radiation Research Institute, SPring-8, 1-1-1 Kouto, Sayo, Hyogo 679-5198, Japan. [4]Research Center for Quantum Sensing, Zhejiang Lab, 310000 Hangzhou, P. R. China. [5]Photon Science Division, Paul Scherrer Institut, CH-5232 Villigen PSI, Switzerland. ✉ e-mail: ccao@zju.edu.cn; yusong_phys@zju.edu.cn

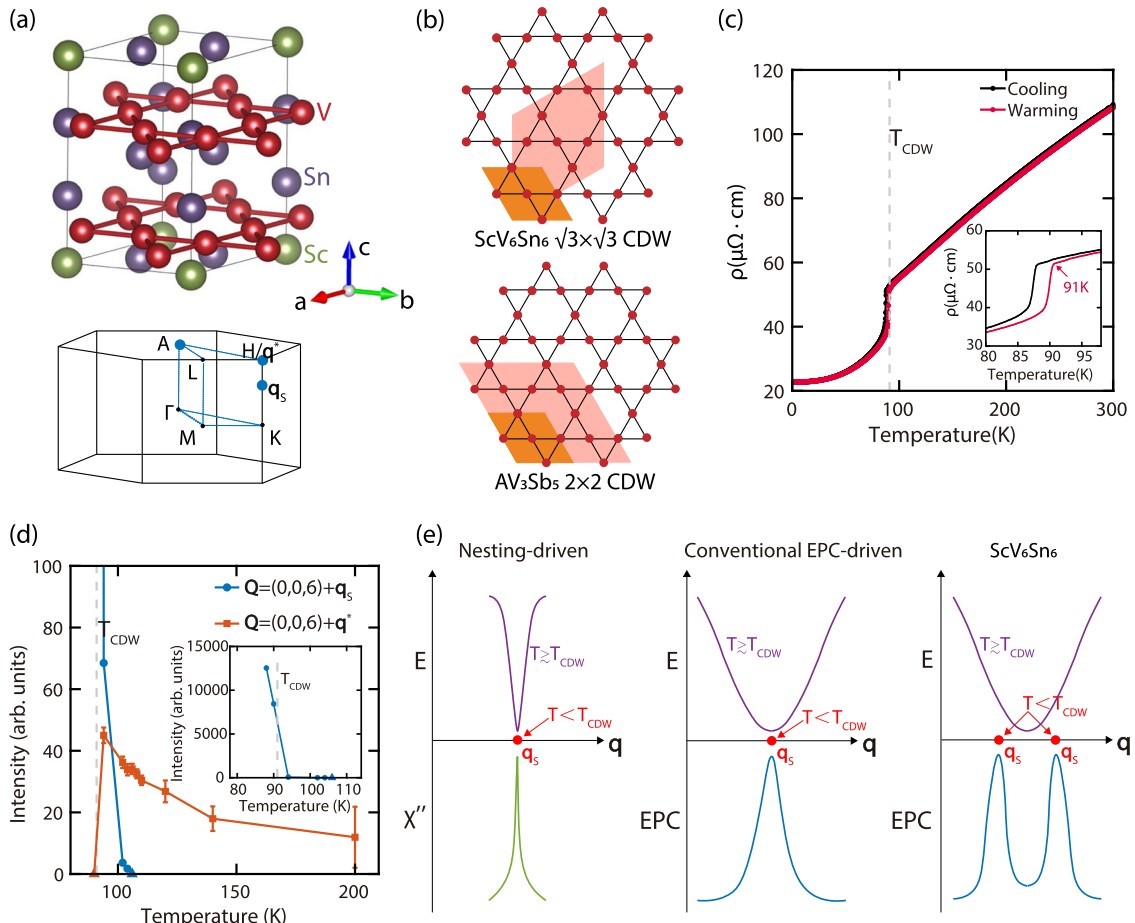

**Fig. 1 | Crystal structure, resistivity of ScV$_6$Sn$_6$, and schematics of its CDW formation. a** Crystal structure of ScV$_6$Sn$_6$[45], visualized using VESTA[70], with the Brillouin zone shown below. The blue circles are **q**-points probed in this work. **b** Expansion of the unit cell in the *ab*-plane for the CDWs in $A$V$_3$Sb$_5$ and ScV$_6$Sn$_6$. **c** The electrical resistivity of ScV$_6$Sn$_6$, the inset zooms in around $T_{CDW}$. **d** A comparison of the integrated intensities for **q**$_s$-CDW and **q**$^*$-CDW, obtained from *l*-scans centered at (0, 0, 6). The inset zooms out to highlight the rapid growth of **q**$_s$-CDW. The triangles represent the absence of a detectable peak. **e** Schematics of the formation process for CDWs of different origins. A nesting-driven CDW is triggered

by a divergent electronic susceptibility $\chi''$, and phonon softening occurs over a small region in **q**-space. A conventional EPC-driven CDW results from a peak in the wavevector-dependent EPC and manifests through phonon softening over an extended region in **q**-space. In these two cases, the peak in $\chi''$ or EPC, the softest phonon mode, and static CDW order, all occur at the same position in **q**-space. In ScV$_6$Sn$_6$, the softest phonon mode occurs at **q**$^*$, whereas static CDW order and the EPC peak occur at a different wavevector **q**$_s$. The error bars in (**d**) are least-square fit errors of 1 s.d.

charge gap formation[47], distinct from $A$V$_3$Sb$_5$[48,49]. This view is reinforced by electronic structure measurements, which in addition identify the lattice or a Lifshitz transition as instrumental for the CDW in ScV$_6$Sn$_6$[50,51].

Phonons play crucial roles in the CDWs of the kagome metals $A$V$_3$Sb$_5$ and FeGe, and understanding their behaviors offered critical insights regarding the mechanism underlying CDW formation[38,52–54]. Whereas CDWs in both the weak- and strong-coupling limits are expected to exhibit soft phonons above the CDW ordering temperature, inelastic X-ray scattering (IXS) measurements of $A$V$_3$Sb$_5$ reveal an absence of such phonon softening, suggesting an unconventional CDW near the van Hove filling[38,53]. Inelastic neutron scattering unveils the hardening of a longitudinal optical phonon inside the CDW state of CsV$_3$Sb$_5$, implicating a key role of electron-phonon coupling (EPC) in the CDW formation[52]. IXS measurements of FeGe uncover a charge dimerization and significant spin-phonon coupling, which intertwine with magnetism to drive the CDW formation[54]. In the case of ScV$_6$Sn$_6$, theoretical calculations find competing lattice instabilities[55] and the softening of a flat phonon mode is observed via a combination of experimental and theoretical techniques[56].

Here, we use IXS to study the lattice dynamics related to CDW formation in ScV$_6$Sn$_6$, revealing a clear phonon softening above the

first-order CDW ordering temperature $T_{CDW} \approx 91$ K [Fig. 1c]. Whereas long-range static CDW order occurs at **q**$_s$ = $(\frac{1}{3}, \frac{1}{3}, \frac{1}{3})$, corresponding to a $\sqrt{3} \times \sqrt{3} \times 3$ CDW (**q**$_s$-CDW), the phonon softening is most prominent at **q**$^*$ = $(\frac{1}{3}, \frac{1}{3}, \frac{1}{2})$, corresponding to a short-range $\sqrt{3} \times \sqrt{3} \times 2$ CDW (**q**$^*$-CDW). **q**$^*$-CDW gains in intensity upon cooling, but becomes suppressed below $T_{CDW}$, replaced by **q**$_s$-CDW via a first-order transition [Fig. 1d]. These observations depict a CDW formation process in ScV$_6$Sn$_6$ distinct from known nesting-driven or typical EPC-driven CDWs[57], with the softest phonon occurring at **q**$^*$, while the static CDW occurs at a distinct wavevector **q**$_s$ [Fig. 1e]. First-principles calculations reveal that although **q**$^*$-CDW is energetically more favorable at the density functional theory level, a **q**-dependent EPC promotes **q**$_s$-CDW as the ground state, and also leads to strong electron scattering above $T_{CDW}$, accounting for the large resistivity drop upon cooling below $T_{CDW}$. These findings underscore the importance of EPC-driven many-body physics in ScV$_6$Sn$_6$ and provide a further example of unconventional CDW on the kagome lattice.

## Results
### Competition between two distinct CDWs
Elastic scattering in ScV$_6$Sn$_6$ was measured by setting the energy transfer in IXS to zero, with results presented in Fig. 2. For $T \gtrsim 100$ K,

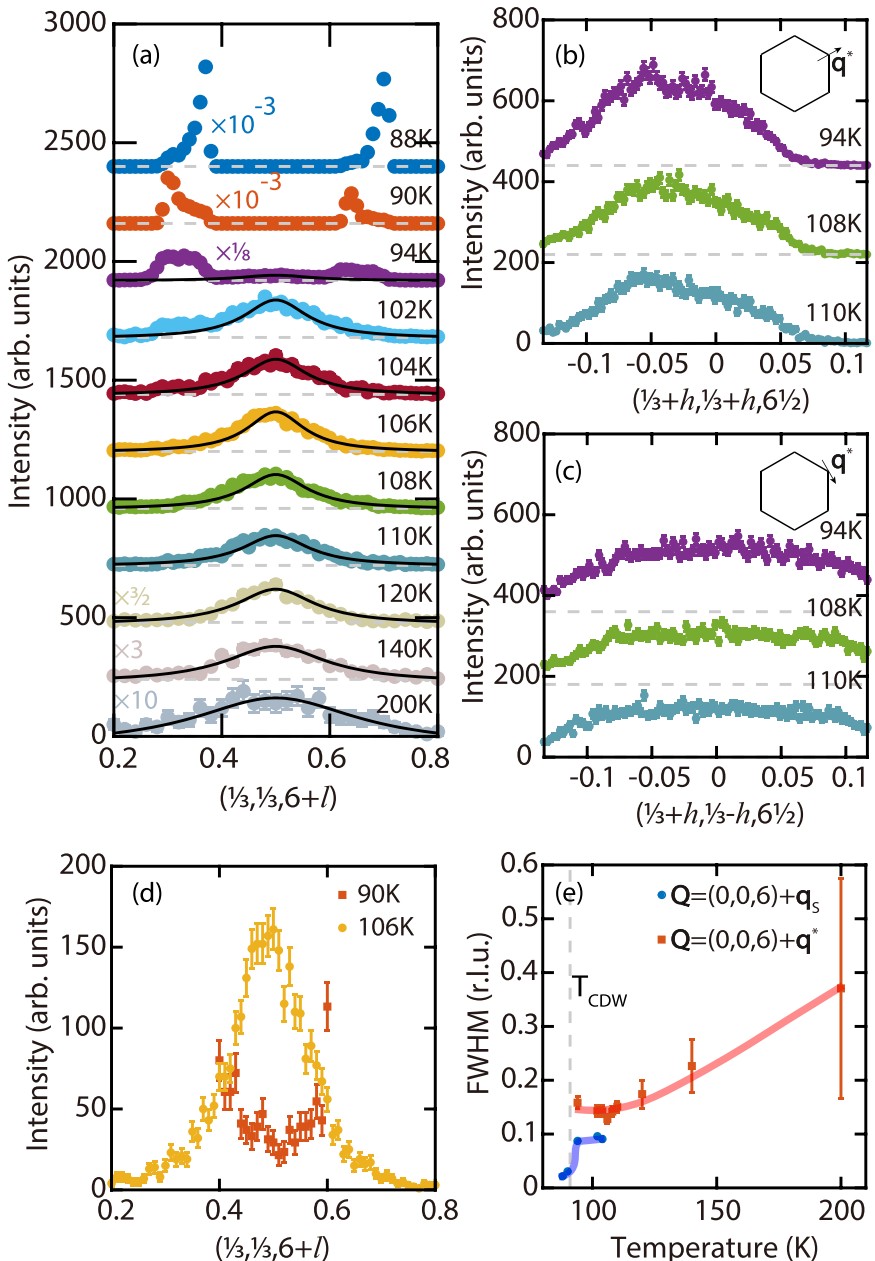

**Fig. 2 | Competing CDWs in ScV₆Sn₆.** Elastic scans along **a** $(\frac{1}{3},\frac{1}{3},6+l)$, **b** $(\frac{1}{3}+h,\frac{1}{3}+h,6\frac{1}{2})$, and **c** $(\frac{1}{3}+h,\frac{1}{3}-h,6\frac{1}{2})$, measured at various temperatures. The data have been shifted vertically (dashed gray lines) for clarity. The solid black lines in (**a**) are fits to the Lorentzian function. **d** A comparison between elastic scans along $(\frac{1}{3},\frac{1}{3},6+l)$ between 90 K and 106 K. The scan at 90 K is limited in range as the scattering around $l=\frac{1}{3}$ and $\frac{2}{3}$ are too intense, due to less beam attenuation

compared to the 90 K scan in panel (**a**). The 106 K scan is identical to that in panel (**a**). The different attenuation used in different scans have been corrected for, see "Methods". **e** The FWHM along $l$ for $\mathbf{q}_s$-CDW and $\mathbf{q}^*$-CDW as a function of temperature. The thick lines are guides to the eye. The error bars in (**a–d**) represent statistical errors of 1 s.d., and the error bars in (**e**) are least-square fit errors of 1 s.d.

clear diffuse scattering centered around $(0, 0, 6) + \mathbf{q}^*$ are observed in $l$-scans [Fig. 2a]. Scans along $(\frac{1}{3}+h,\frac{1}{3}+h,6\frac{1}{2})$ and $(\frac{1}{3}+h,\frac{1}{3}-h,6\frac{1}{2})$ confirm the short-range nature of these peaks along two orthogonal in-plane directions [Fig. 2b, c]. The $\mathbf{q}^*$-CDW peak is significantly broader in the $hk$-plane than along $l$, and the peak asymmetry in Fig. 2b likely results from the variation in structure factors of the associated soft phonons in different Brillouin zones, since $\mathbf{q}^*$ ($H$) is a high-symmetry point and the energies and damping rates of phonon modes should be symmetric around it (Supplementary Note 1 and Supplementary Fig. 1). These diffuse scattering centered around $(0, 0, 6) + \mathbf{q}^*$ evidence an unreported $\mathbf{q}^*$-CDW in ScV₆Sn₆, distinct from $\mathbf{q}_s$-CDW in its ground state[45]. As the temperature is lowered, a weak peak around

$(0, 0, 6) + \mathbf{q}_s$ is first observed at 104 K and quickly gains in intensity upon further cooling. In contrast, the $\mathbf{q}^*$-CDW peak at $\mathbf{q}^*$ is no longer discernible at 90 K [Fig. 2d]. The temperature evolution of the integrated intensities are compared for $\mathbf{q}_s$-CDW and $\mathbf{q}^*$-CDW in Fig. 1d, clearly revealing their competition. At $T = 88$ K (below $T_{CDW}$), the peak intensity of $\mathbf{q}_s$-CDW is at least 3 orders of magnitude larger than the maximum peak intensity of $\mathbf{q}^*$-CDW (occurring at $\approx 94$ K), accounting for why only $\mathbf{q}_s$-CDW was detected in lab source X-ray diffraction measurements[45].

The full-widths at half-maximum (FWHM) of the measured CDW peaks along $l$ are compared in Fig. 2e, revealing that $\mathbf{q}^*$-CDW remains short-range down to 94 K. By fitting the Lorentzian function to $l$-scans

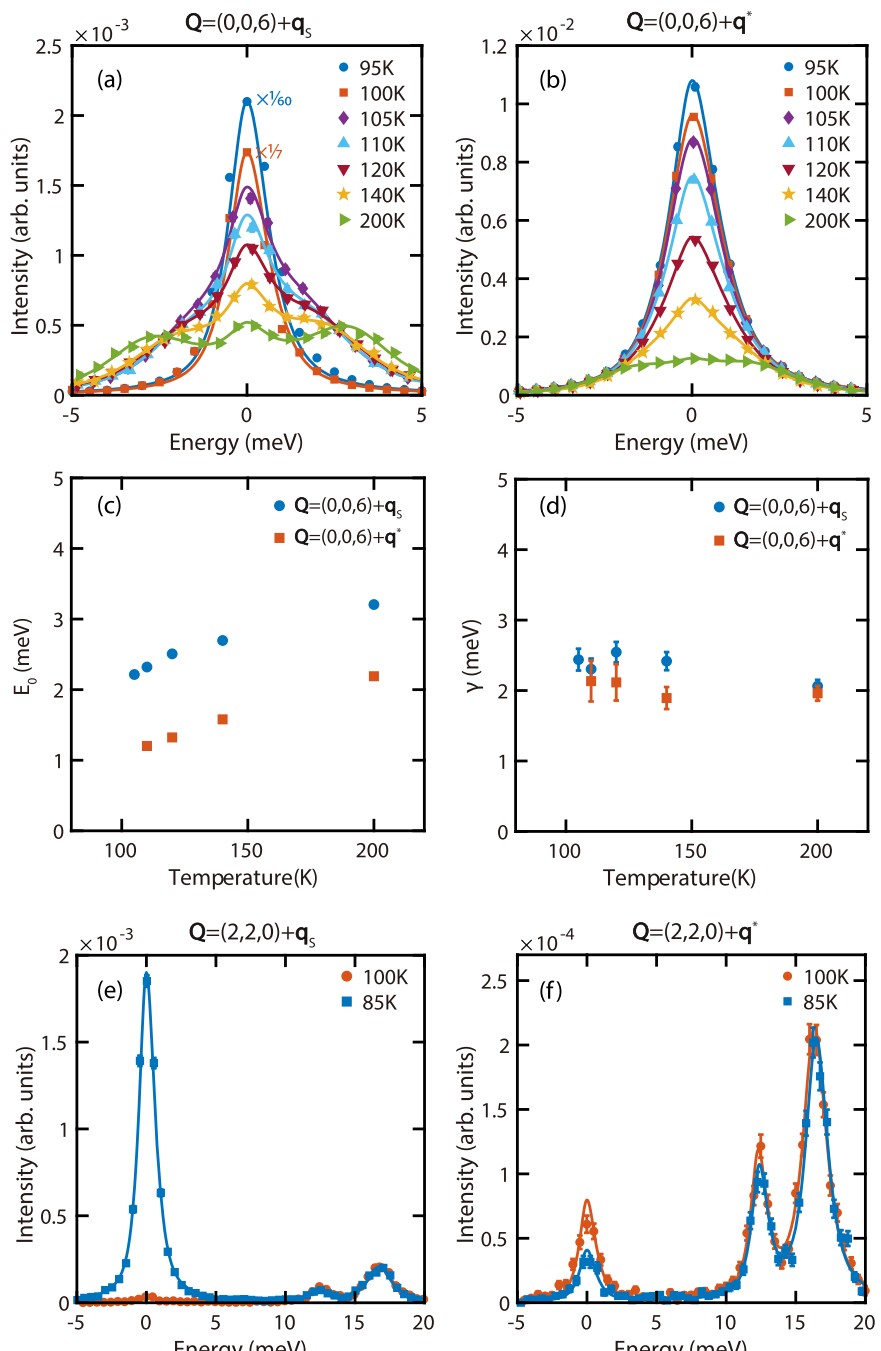

**Fig. 3 | Measurements of lattice dynamics in $ScV_6Sn_6$.** IXS spectra at **a** $(0, 0, 6) + \mathbf{q}_s$ and **b** $(0, 0, 6) + \mathbf{q}^*$, measured at various temperatures. The solids lines are fits to a DHO and an elastic peak, convolved with the instrumental resolution. From these fits, **c** $E_0$ and **d** $\gamma$ are extracted and compared between the two wavevectors. The 95 K and 100 K scans in (**a**) are resolution-limited. The 95 K, 100 K, and 105 K data in (**b**) contain an inelastic response but cannot be reliably distinguished from the elastic peak. DHO fit parameters are not shown for these scans. IXS spectra at **e** $(2, 2, 0) + \mathbf{q}_s$ and **f** $(2, 2, 0) + \mathbf{q}^*$, compared between 85 K and 100 K. The solids lines are fits to DHOs and an elastic peak, convolved with the instrumental resolution. The error bars in (**a**), (**b**), (**e**) and (**f**) represent statistical errors of 1 s.d., and the error bars in (**c**) and (**d**) are least-square fit errors of 1 s.d.

of $\mathbf{q}^*$-CDW, we find the extracted correlation lengths is around 20 Å for $T \lesssim 110$ K. In the case of $\mathbf{q}_s$-CDW, the associated peaks are also broad for $T \gtrsim 100$ K, but sharpen for $T \lesssim 90$ K, with a correlation length exceeding 100 Å. We note the peaks associated with $\mathbf{q}_s$-CDW in Fig. 2a appear slightly away from $\mathbf{q}_s$ in some measurements, which may result from a distribution of short-range $\mathbf{q}_s$-CDW clusters, domain formation due to the lowering of lattice symmetry below $T_{CDW}$, or a small sample misalignment.

## Lattice dynamics associated with the formation of CDWs

To probe the lattice dynamics associated with the CDW formation in $ScV_6Sn_6$, IXS measurements were carried out at $(0, 0, 6) + \mathbf{q}_s$ and $(0, 0, 6) + \mathbf{q}^*$ [Fig. 3a, b], clearly revealing soft phonons at both positions. Whereas the soft phonons form two peaks centered around the elastic line at $T = 200$ K, they further soften upon cooling and form a single quasielastic peak. To quantitatively analyze the phonon spectra, the phonon contributions in Fig. 3a, b are fit using the general damped

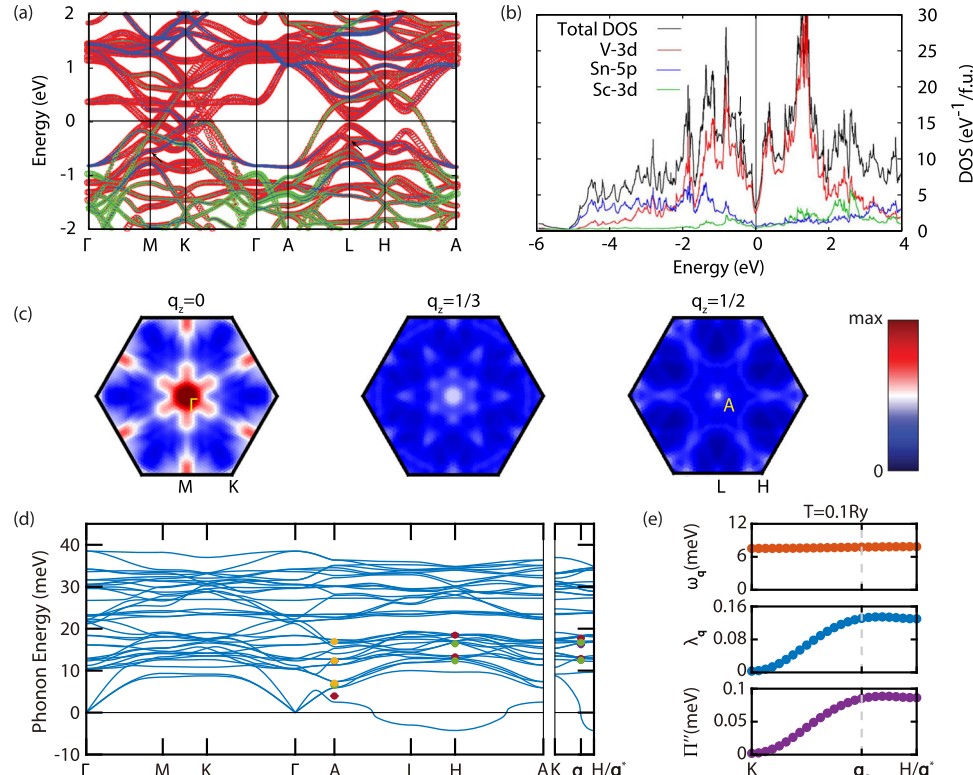

**Fig. 4 | First-principles calculations in ScV$_6$Sn$_6$. a** Calculated electronic band structure of ScV$_6$Sn$_6$ with orbital characters, in the absence of spin-orbit interaction. Red corresponds to V-3$d$ orbitals, blue Sc-3$d$ orbitals, and green Sn-5$p$ orbitals. The size of the circles represents the corresponding orbital weights. **b** Total and projected density of states (DOS), V-3$d$ orbitals dominate within $E_F \pm 1$ eV. **c** Nesting functions calculated at $q_z = 0$, 1/3 and 1/2. **d** The phonon spectrum of ScV$_6$Sn$_6$ calculated using DFPT. The red, green, purple and yellow dots respectively represent phonon modes detected in (006), (220), (113) and (300) Brillouin zones. The error bars are the standard deviations of phonon energies at various temperatures. **e** Calculated phonon dispersion (top), EPC strength $\lambda_{\mathbf{q}\nu}$ (middle), and phonon self-energy $\Pi''_{\mathbf{q}\nu}$ (bottom) for the soft phonon mode ($\nu = 1$) along $K - H$, at a high electron temperature ($T = 0.1$ Ry). See Supplementary Note 4 and Supplementary Fig. 7 for a comparison with calculations at $T = 0.01$ Ry.

harmonic oscillator (DHO)[58,59]:

$$S(E) = \frac{A}{1 - \exp\left(-\frac{E}{k_B T}\right)} \frac{2}{\pi} \frac{\gamma E}{\left(E^2 - E_0^2\right)^2 + (E\gamma)^2}, \qquad (1)$$

shown as solid lines. In the DHO model, $A$ is an intensity scale factor, $E_0$ is the undamped phonon energy, and $\gamma$ is the damping rate (peak FWHM when $\gamma \ll E_0$). The fit values of $E_0$ decrease markedly with cooling for both $\mathbf{q}_s$ and $\mathbf{q}^*$, with the phonons at $\mathbf{q}^*$ softer than those at $\mathbf{q}_s$ [Fig. 3c]. In contrast, the damping rate $\gamma$ changes relatively little with temperature, with the phonons at $\mathbf{q}_s$ slightly more strongly damped than those at $\mathbf{q}^*$. The observation of phonon softening in tandem with the growth of $\mathbf{q}^*$-CDW suggests it is dynamic in nature, and the diffuse character of $\mathbf{q}^*$-CDW is a result of softening over an extended region in momentum space. On the other hand, while $\mathbf{q}_s$-CDW develops at $T = 104$ K in the elastic channel [Fig. 2a], the corresponding $\mathbf{q}_s$ phonon mode retains well-defined energy at a similar temperature (105 K), indicating $\mathbf{q}_s$-CDW develops via the growth of an elastic central peak, rather than phonons softening to zero energy. The short-range $\mathbf{q}_s$-CDW precursors detected at $T \gtrsim 100$ K [Fig. 2a, e] suggest the first-order transition at $T_{CDW}$ is likely order-disorder type, as suggested for $A$V$_3$Sb$_5$[53].

IXS measurements at $\mathbf{q}_s$ and $\mathbf{q}^*$ were also carried out in the (220) Brillouin zone [Fig. 3e, f], which is dominated by phonons polarized in the $ab$-plane. In contrast, measurements in the (006) Brillouin zone are dominated by $c$-axis polarized phonons. For both $\mathbf{q}_s$ and $\mathbf{q}^*$, soft phonons are hardly detectable in the (220) Brillouin zone, although the

presence of $\mathbf{q}^*$-CDW is evidenced by the more intense elastic peak at 100 K relative to 85 K. For comparison, the elastic peak at $\mathbf{q}_s$ gains in intensity upon cooling from 100 K to 85 K, due to the appearance of $\mathbf{q}_s$-CDW. The opposing temperature evolution of elastic peaks in Fig. 3e, f are consistent with the competition between $\mathbf{q}_s$-CDW and $\mathbf{q}^*$-CDW revealed in Fig. 2. The much weaker soft phonons in the (220) Brillouin zone suggest $\mathbf{q}^*$-CDW is associated with dominantly $c$-axis polarized lattice vibrations, similar to $\mathbf{q}_s$-CDW which is mostly due to Sc and Sn displacements along the $c$-axis[45]. Two additional phonon branches are also detected in Fig. 3e, f, with phonon energies at $\mathbf{q}_s$ slightly higher than those at $\mathbf{q}^*$. The fact these phonons hardly change across $T_{CDW}$ suggests they are likely associated with in-plane vibrations of the lattice. Additional phonon modes that do not change significantly across $T_{CDW}$ are also detected in several Brillouin zones (see Supplementary Note 2 and Supplementary Fig. 2), the energies of these phonon modes are shown in Fig. 4d and Supplementary Fig. 3.

**First-principles calculations**

First-principles calculations were employed to understand the experimentally observed CDWs in ScV$_6$Sn$_6$, with the calculated electronic structure shown in Fig. 4a. We find the electronic structure close to the Fermi level is dominated by V-3$d$ orbitals, which can also be seen in the projected density of states (DOS) [Fig. 4b], in agreement with the previous study[55]. Characteristic features of the kagome lattice are identified in the electronic structure, including Dirac cones at $K$ ($\sim -0.1$ eV and $-0.04$ eV) and $H$ ($\sim -0.5$ eV), and topological flat bands around $-0.5$ eV at $M$ and $L$, which manifest as van Hove-like features around $-0.5$ eV in the electronic DOS [Fig. 4b].

To probe the origins of the competing CDWs in $ScV_6Sn_6$, the nesting function $J(\mathbf{q}) = \frac{1}{N_k} \sum_{\nu,\mu,\mathbf{k}} \delta(\epsilon_{\mu\mathbf{k}}) \delta(\epsilon_{\nu\mathbf{k+q}})$ is computed, where $\epsilon_{\mu\mathbf{k}}$ is the energy (with respect to the Fermi energy) of band $\mu$ at $\mathbf{k}$. As can be seen in Fig. 4c, the most prominent feature of $J(\mathbf{q})$ is at the $M$-point, which does not correspond to a CDW instability [Fig. 4d], and multiple marginal features are observed in the $q_z = \frac{1}{3}$ and $\frac{1}{2}$ planes [Fig. 4c]. Most importantly, in the $q_z = \frac{1}{3}$ plane, no peak is present at $\mathbf{q}_s = (\frac{1}{3}, \frac{1}{3}, \frac{1}{3})$, suggesting that Fermi-surface nesting is completely irrelevant in the formation of $\mathbf{q}_s$-CDW, consistent with previous findings[47,55]. In the $q_z = \frac{1}{2}$ plane, hot spots are found around $(\frac{1}{6}, \frac{1}{6}, \frac{1}{2})$ and $\mathbf{q}^* = (\frac{1}{3}, \frac{1}{3}, \frac{1}{2})$, indicating a possible contribution of nesting towards $\mathbf{q}^*$-CDW. The results in Fig. 4a–c are obtained without spin-orbit coupling (SOC), and adding SOC leads to only marginal changes (Supplementary Note 3 and Supplementary Figs. 4 and 5).

In addition to Fermi-surface nesting, EPC can also drive a CDW transition. To elucidate the role of phonons in the competing CDWs of $ScV_6Sn_6$, we calculated its phonon spectrum using DFPT[60], shown in Fig. 4d. The calculations reproduce several experimentally measured phonons modes at $A$ and $H$ [circles in Fig. 4d], demonstrating consistency between theory and experiment. In particular, several calculated phonon modes are nearly degenerate around 12.8 meV and 16.8 meV at $H$ ($\mathbf{q}^*$), as well as around 13.0 meV and 17.0 meV at $\mathbf{q}_s$. These phonons match the experimental observations in Fig. 3e, f, and are dominated by the in-plane motion of Sn atoms. Similar to previous calculations[55], imaginary phonon modes are present along $A-L-H$, with the imaginary $A_1$ mode at $H$ lowest in energy. In addition to the soft phonons experimentally observed at $\mathbf{q}_s$ and $\mathbf{q}^*$, a low energy ~ 4.0 meV phonon mode without softening is detected experimentally at $A = (0, 0, \frac{1}{2})$, occurring at a higher energy than the calculated ~2.5 meV mode [Fig. 4d]. Furthermore, two new phonon modes are identified experimentally at $A$ upon entering the $\mathbf{q}_s$-CDW state (Supplementary Note 2 and Supplementary Figs. 2 and 3).

In most cases, the imaginary phonon mode with the lowest energy would drive a CDW transition, which is clearly not the case in $ScV_6Sn_6$, since static CDW occurs at $\mathbf{q}_s$, rather than at $\mathbf{q}^*$ ($H$) which has the lowest phonon mode. More surprisingly, our calculations indicate an absence of imaginary phonons at $\mathbf{q}_s$ [Fig. 4d and Supplementary Fig. 7], suggesting that at the level of density functional theory, $\mathbf{q}_s$-CDW is also less competitive than the undistorted $P6/mmm$ structure. This is reflected in the recovery of the undistorted structure when relaxing the supercell modulated by the lowest energy phonon mode at $\mathbf{q}_s$ (see "Methods").

To address this problem, we calculated the phonon self-energy $\Pi''_{\mathbf{q}\nu}$ (proportional to the phonon peak width in energy) and $\mathbf{q}$-dependent EPC strength $\lambda_{\mathbf{q}\nu}$ for the lowest phonon mode ($\nu = 1$) along $K$-$H$ at different electron temperatures (see "Methods" and Supplementary Note 4), which are related to the EPC matrices $g_{mn}^{\nu}(\mathbf{k},\mathbf{q})$ via[61,62]:

$$\Pi''_{\mathbf{q}\nu} = \mathrm{Im}\left[\sum_{mn\mathbf{k}} |g_{mn}^{\nu}(\mathbf{k},\mathbf{q})|^2 \frac{f_{n\mathbf{k}} - f_{m\mathbf{k+q}}}{\epsilon_{n\mathbf{k}} - \epsilon_{m\mathbf{k+q}} - \omega_{\mathbf{q}\nu} + i\eta}\right], \quad (2)$$

and

$$\lambda_{\mathbf{q}\nu} = \frac{1}{N_F \omega_{\mathbf{q}\nu}} \sum_{mn\mathbf{k}} |g_{mn}^{\nu}(\mathbf{k},\mathbf{q})|^2 \delta(\epsilon_{n\mathbf{k}}) \delta(\epsilon_{m\mathbf{k+q}}). \quad (3)$$

We find that at a high electron temperature (~0.1 Ry), the $P6/mmm$ structure of $ScV_6Sn_6$ is stable without imaginary phonons [top of Fig. 4e and Supplementary Fig. 6], while both $\Pi''_{\mathbf{q}}$ and $\lambda_{\mathbf{q}}$ for the lowest phonon mode exhibit humps around $\mathbf{q}_s$ [middle and bottom of Fig. 4e], evidencing a $\mathbf{q}$-dependent EPC. This is further corroborated by calculations at a low electron temperature (~0.01 Ry), where the hump in $\Pi''_{\mathbf{q}}$ becomes further enhanced [Supplementary Note 4 and Supplementary Fig. 7]. In combination with the absence of features in the

nesting function [Fig. 4c], these results suggest that the $\mathbf{q}$-dependent EPC plays a key role in selecting $\mathbf{q}_s$-CDW as the ground state in $ScV_6Sn_6$.

## Discussion

CDWs usually occur via phonon softening, corresponding to coherent lattice oscillations that gradually become more competitive in energy, or the growth of a central peak that reflects the ordering of local CDW patches. The development of CDWs in one dimension as modeled by Peierls[57], and in two-dimensional systems such as $2H$-$NbSe_2$[63] and $BaNi_2As_2$[64,65], are accompanied by prominent phonon softening. While such phonon softening is limited to a small range in momentum in Peierls' model, it occurs over an extended range in $2H$-$NbSe_2$ and $BaNi_2As_2$, similar to the observed behavior of $\mathbf{q}^*$-CDW in $ScV_6Sn_6$. On the other hand, order-disorder CDW transitions have been reported in systems such as $(Ca_{1-x}Sr_x)_3Rh_4Sn_{13}$[66] and $AV_3Sb_5$[53], and likely characterize the formation of $\mathbf{q}_s$-CDW in $ScV_6Sn_6$. Thus, the CDW formation process in $ScV_6Sn_6$ is unique in that both prominent phonon softening and the growth of a central peak are observed, with the two effects associated with different wavevectors, a result of competing CDW instabilities.

There are three implications that directly result from our experiments and first-principles calculations. First, while $\mathbf{q}^*$-CDW is energetically favored in DFT calculations, $\mathbf{q}_s$-CDW is the ground state of $ScV_6Sn_6$. This apparent inconsistency could result from a $\mathbf{q}$-dependent EPC selecting $\mathbf{q}_s$-CDW as the ground state. This is because the calculated electronic states and phonon energies are "bare" particles, without full consideration of EPC, which leads to considerable electron/phonon self-energies in the strong-coupling limit. We argue that if the phonon-induced electronic self-energy is properly taken into consideration in many-body theories beyond DFT, $\mathbf{q}_s$-CDW should become energetically more competitive than $\mathbf{q}^*$-CDW. This view of a $\mathbf{q}$-dependent EPC favoring $\mathbf{q}_s$-CDW as the ground state is supported by an enhancement of the phonon self-energy $\Pi''_{\mathbf{q}\nu}$ around $\mathbf{q}_s$ upon decreasing the electron temperature (Supplementary Note 4 and Supplementary Fig. 7). Second, both $\mathbf{q}_s$-CDW and $\mathbf{q}^*$-CDW are associated with the $A_1$ phonon mode, for which the V kagome lattice is mostly unaffected. Since the electronic states near the Fermi level are dominated by the V-$3d$ orbitals, gap-opening associated with either $\mathbf{q}_s$-CDW or $\mathbf{q}^*$-CDW is unlikely to be prominent in $ScV_6Sn_6$. Third, our findings explain the substantial drop in resistivity below $T_{CDW}$ [Fig. 1c]: the $\mathbf{q}$-dependent EPC and extended phonon softening revealed in this work both enhance electron scattering above $T_{CDW}$, and the removal of these effects below $T_{CDW}$ strongly reduces electron scattering, consistent with optical conductivity measurements[47].

Furthermore, it is interesting to consider whether the competition between CDW instabilities in $ScV_6Sn_6$ could be tilted in favor of $\mathbf{q}^*$-CDW via external tuning. In this regard, electrical transport measurements in pressurized $ScV_6Sn_6$ reveal that the sharp drop in resistivity associated with $\mathbf{q}_s$-CDW persists up to ~2.0 GPa, beyond which it is suddenly replaced by a much weaker kink in resistivity, before becoming fully suppressed at ~2.4 GPa[46]. The sudden qualitative change in resistivity anomaly above ~2.0 GPa is suggestive of a change in the ground state, and the much less pronounced resistivity anomaly between ~2.0 GPa and ~2.4 GPa suggests the associated transition being second-order. In such a scenario, a distinct possibility is that $\mathbf{q}^*$-CDW becomes the ground state between ~2.0 GPa and ~2.4 GPa, and since $\mathbf{q}^*$-CDW develops through phonon softening [Fig. 3] as in $2H$-$NbSe_2$[63] and $BaNi_2As_2$[64,65], the corresponding resistivity anomaly would be likewise rather subtle.

In conclusion, we uncovered competing CDW instabilities in the kagome metal $ScV_6Sn_6$, which lead to a unique CDW formation process with the dominant soft phonons and the ground state CDW occurring at different wavevectors, distinct from typical phonon-driven CDWs. The two CDWs develop in highly different manners, suggestive of distinct mechanisms, and differentiate $ScV_6Sn_6$ from CDWs in other

kagome metals. As the $\mathbf{q}_s$-CDW ground state is not captured in first-principles DFT calculations, it is likely a correlated many-body effect driven by a $\mathbf{q}$-dependent EPC. Our findings demonstrate a strong EPC on the kagome lattice could lead to nearly degenerate ground states, a setup primed for the emergence of unusual phases of matter.

## Methods

### Experimental details

Single crystals of $ScV_6Sn_6$ were grown using the self-flux method with Sc:V:Sn = 1:6:40[45]. Distilled dendritic scandium pieces (99.9%), vanadium pieces (99.7%), and tin shot (99.99+%) were placed in an alumina crucible and sealed under vacuum in a quartz ampoule. The ampoule was placed in a furnace and heated to 1150 °C for 12 h, then held at 1150 °C for 20 h. The sample was then cooled to 750 °C at a rate of 1 °C/h, at which point the excess Sn flux was removed with the aid of a centrifuge. The resulting crystals are plate-like with the $c$-axis normal to the plates, and typical sample dimensions are around $1 \times 1 \times 0.5$ mm³. Electrical resistivity was measured with the standard four-point method.

Inelastic X-ray scattering measurements were carried out in the transmission geometry using the BL35XU beamline[67] at SPring-8, Japan. The incident photon energy is 21.7476 keV. A ~70-μm-thick sample [Supplementary Fig. 8a], comparable to the attenuation length of ~20 keV X-rays in $ScV_6Sn_6$, was prepared and mounted on a Cu post using silver epoxy. The instrumental energy resolution was measured using a piece of polymethyl methacrylate (PMMA) and parametrized using a pseudo-Voigt function. The instrumental resolution function $R(E)$ is then obtained by normalizing the pseudo-Voigt function to unit area [Supplementary Fig. 8b]. The full-width at half-maximum (FWHM) of $R(E)$ is ~1.38 meV. For temperatures around $T_{CDW}$, measurements were consistently carried out upon warming. Momentum transfer is referenced in reciprocal lattice units, using the high-temperature hexagonal $P6/mmm$ cell of $ScV_6Sn_6$, with $a = b \approx 5.47$ Å and $c \approx 9.16$ Å[45]. All measured scattering intensities are normalized by a monitor right before the sample. For momentum scans of elastic scattering, an attenuator was used to avoid saturating the detector when needed, which can be corrected for via the calibrated attenuation of the attenuator. These corrections have been applied to the data in Fig. 2.

### Analysis of the experimental data

The elastic scans in momentum around $\mathbf{q}^*$ are fit to a Lorentzian function:

$$I(\mathbf{Q}) = b + \frac{A}{\pi} \frac{\frac{\Gamma}{2}}{(\mathbf{Q} - \mathbf{Q}^*)^2 + \left(\frac{\Gamma}{2}\right)^2}, \quad (4)$$

where $b$ is a small constant, $A$ is the integrated area, $\mathbf{Q}^* = (0, 0, 6) + \mathbf{q}^*$ is the center of peak, and $\Gamma$ is the full-width at half-maximum (FWHM). For temperatures with detectable $\mathbf{q}_s$-CDW intensity, regions around $(\frac{1}{3}, \frac{1}{3}, 6\frac{1}{3})$ and $(\frac{1}{3}, \frac{1}{3}, 6\frac{2}{3})$ are masked in the fit. Integrated intensities and FWHMs for $\mathbf{q}_s$-CDW are then determined numerically from the data around $(\frac{1}{3}, \frac{1}{3}, 6\frac{1}{3})$, after subtracting the fit to the Lorentzian function. For temperatures without detectable $\mathbf{q}^*$-CDW, the integrated intensities and FWHMs for $\mathbf{q}_s$-CDW are likewise obtained numerically, after subtracting a small constant term determined from the mean of data points away from the $\mathbf{q}^*$-CDW peaks. The obtained integrated intensities and FWHMs for $\mathbf{q}_s$-CDW and $\mathbf{q}^*$-CDW are shown in Figs. 1d and 2e.

Using the least squares method, all measured experimental phonon intensities are fit to the expression:

$$I(E) = b + cR(E - \delta E) + \sum_{i=1}^{n} \int_{-\infty}^{\infty} [S_i(E - \delta E - E')]R(E')dE', \quad (5)$$

where $n$ is the minimum number of phonon modes that capture the experimentally measured data, and the integrals correspond to convolutions with the instrumental resolution $R(E)$. In practice, since $R(E)$ has a FWHM of ~1.38 meV, the integrals are numerically carried out in the energy range $[-20, 20]$ meV. The above equation contains a small constant term $b$, a resolution-limited elastic peak $cR(E)$ and $n$ general damped harmonic oscillators (DHOs)[58,59], with each phonon mode represented by the DHO $S_i(E)$:

$$S_i(E) = \frac{A_i}{1 - \exp\left(-\frac{E}{k_B T}\right)} \frac{2}{\pi} \frac{\gamma_i E}{(E^2 - E_{0i}^2)^2 + (E\gamma_i)^2}, \quad (6)$$

where $A_i$ is the intensity scale factor, $E_{0i}$ is the phonon energy in the absence of damping, and $\gamma_i$ is the damping rate, all for phonon mode $i$. In the limit of $\gamma_i \to 0$ (or the phonon mode is resolution-limited), the above equation for $S_i(E)$ can be replaced by:

$$S_i(E) = \frac{A_i}{1 - \exp\left(-\frac{E}{k_B T}\right)} \frac{\delta(E - E_{0i}) - \delta(E + E_{0i})}{E}. \quad (7)$$

In addition, the difference between the actual zero energy and the nominal zero energy is contained in our model as a free parameter $\delta E$ to account for shifts in energy between different scans. Possible shifts of energy within each scan are considered to be negligible and ignored in our analysis. The data in Fig. 3 and Supplementary Fig. 2 have been shifted by $\delta E$ obtained in the fits.

As the temperature is cooled and $\mathbf{q}_s$-CDW develops, phonons at $\mathbf{q}_s$ become difficult to measure due to the elastic tail of the $\mathbf{q}_s$-CDW peak. A result of this is that the 95 K and 100 K data become almost resolution-limited around the elastic line, and no phonons are contained in the corresponding fits [Fig. 3a and Supplementary Fig. 9a]. On the other hand, while the soft phonon mode at $\mathbf{q}^*$ becomes a single peak centered around the elastic peak at 95 K, 100 K, and 105 K, they are broader in energy than the instrumental resolution [Fig. 3b and Supplementary Fig. 9b]. Although these soft phonons can be described by the DHO model, it is not possible to reliably extract $E_{0i}$ and $\gamma_i$.

### First-principles calculations

Electronic structure calculations were carried out using density functional theory (DFT) implemented in Quantum Espresso[68]. The exchange-correlations function was taken within the generalized gradient approximation (GGA) in the parameterization of Perdew, Burke and Ernzerhof[69]. The energy cutoff of plane-wave basis was up to 64 Ry (720 Ry for the augmentation charge). The $3s$, $3p$, $3d$, and $4s$ electrons for Sc and V atoms and $4d$, $5s$, and $5p$ electrons for Sn are considered valence electrons in the employed pseudopotentials. For the undistorted structures, the charge density was calculated self-consistently with a $\Gamma$-centered $12 \times 12 \times 6$ $\mathbf{k}$-point mesh with a Gaussian broadening of 0.01 Ry (low electron temperature). The lattice constants and atomic coordinates were fully relaxed until the force on each atom was less than 1 meV/Å and the internal stress less than 0.1 kbar.

The bare electronic susceptibility was calculated with the Lindhard formula:

$$\chi_0(\omega, \mathbf{q}) = -\frac{1}{N_\mathbf{k}} \sum_{\mu\nu\mathbf{k}} \frac{f(\varepsilon_{\nu\mathbf{k}+\mathbf{q}}) - f(\varepsilon_{\mu\mathbf{k}})}{\omega + \varepsilon_{\nu\mathbf{k}+\mathbf{q}} - \varepsilon_{\mu\mathbf{k}} + i0^+}, \quad (8)$$

where $\mu$, $\nu$ are band indexes, $\varepsilon_{\mu\mathbf{k}}$ is the energy eigenvalue of band $\mu$ at $\mathbf{k}$, $f(\varepsilon_{\mu\mathbf{k}})$ is the Fermi-Dirac distribution.

The imaginary part of the bare electron susceptibility ($\chi_0''(\omega, \mathbf{q})$) is related to the nesting function $J(\mathbf{q})$ through:

$$J(\mathbf{q}) = \lim_{\omega \to 0} \frac{\chi_0''(\omega)}{\omega} = \frac{1}{N_\mathbf{k}} \sum_{\nu,\mu,\mathbf{k}} \delta(\epsilon_{\mu\mathbf{k}}) \delta(\epsilon_{\nu\mathbf{k}+\mathbf{q}}). \quad (9)$$

The phonon spectrum is calculated using density functional perturbation theory (DFPT)[60] on a $4 \times 4 \times 3\mathbf{q}$-grid. The electron-phonon coupling strength $\lambda_{\mathbf{q}\nu}$ and phonon self-energy $\Pi''_{\mathbf{q}\nu}$ were calculated on a $48 \times 48 \times 24$ Wannier-interpolated $\mathbf{k}$-grid using the `EPW` package[62]. Bands derived from Sc-$3d$, V-$3d$ and Sb-$5p$ orbitals from DFT calculations were fit to a tight-binding Hamiltonian using the Maximally Projected Wannier Functions method [Supplementary Fig. 10], which were then used in the `EPW` calculations[61].

We have also performed calculations for the distorted structures associated with $\mathbf{q}^*$- and $\mathbf{q}_s$-CDWs. For $\mathbf{q}_s$-CDW, the initial structure was obtained by imposing the lowest energy phonon mode modulation on a rhombohedral supercell with lattice vectors $\mathbf{A}'_1 = \mathbf{A}_1 + \mathbf{A}_3$, $\mathbf{A}'_2 = \mathbf{A}_2 + \mathbf{A}_3$ and $\mathbf{A}'_3 = -(\mathbf{A}_1 + \mathbf{A}_2) + \mathbf{A}_3$, where $\mathbf{A}_i$ are the lattice vectors for the undistorted $P6/mmm$ structure ($\mathbf{A}_3 \perp \mathbf{A}_{1,2}$ and $\angle(\mathbf{A}_1, \mathbf{A}_2) = 120°$). For $\mathbf{q}^*$-CDW, the initial structure was obtained by imposing the lowest energy phonon mode ($A_1$) modulation on a $3 \times 3 \times 2$ hexagonal supercell with lattice vectors $\mathbf{A}''_1 = 2\mathbf{A}_1 + \mathbf{A}_2$, $\mathbf{A}''_2 = \mathbf{A}_1 + 2\mathbf{A}_2$ and $\mathbf{A}''_3 = 2\mathbf{A}_3$. See Supplementary Fig. 11 for a comparison between primitive unit cells for the undistorted $P6/mmm$ structure and the distorted structures. The initial structures were then fully relaxed so that the force on each atom was less than 1 meV/Å and the internal stress less than 0.1 kbar. The fully relaxed $\mathbf{q}^*$-structure is -7.5 meV/f.u. lower in energy than the undistorted structure, whereas the $\mathbf{q}_s$-structure relaxes back to the undistorted $P6/mmm$ structure, consistent with our calculations that show an absence of imaginary phonon at $\mathbf{q}_s$.

The high electron temperature phonon calculations were simulated with a larger Gaussian smearing of 0.1 Ry (Fig. 4e and Supplementary Figs. 6 and 7)[65].

## Data availability
The IXS data generated in this study are provided in the Source Data file. Source data are provided with this paper.

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

## Acknowledgements

Y.S. and C.C. acknowledge support from the National Key R&D Program of China (No. 2022YFA1402200), the Pioneer and Leading Goose R&D Program of Zhejiang (2022SDXHDX0005), the Key R&D Program of Zhejiang Province, China (2021C01002), and the National Natural Science Foundation of China (No. 12274363, 12274364). Y.L. acknowledges support from the Fundamental Research Funds for the Central Universities (Grant No. 2021FZZX001-03). Measurements at the BL35XU of SPring-8 were performed with the approval of JASRI (Proposal No. 2022B1283). The calculations were performed on the HPC facility at the Center for Correlated Matter, and partially on the HPCC at Hangzhou Normal University.

## Author contributions

Y.S. and C.C. conceived and led the project. S.C. prepared the samples. S.C., H.F. and T.M. carried out the experiments. S.C. and Y.S. carried out the data analysis. C.X., Y.D. and C.C. carried out the first-principles calculations. All authors discussed and interpreted the results. Y.S., C.C., Y.L. and M.S. wrote the paper with input from all authors.

## Competing interests

The authors declare no competing interests.
