## [Peer Review File · Nature Communications]

REVIEWER COMMENTS

Reviewer #1 (Remarks to the Author):

By measuring the inelastic X-ray scattering (IXS) and performing first-principles calculations, , The authors unveiled therein competing CDW instabilities in kagome metal ScV6Sn6, which leads to unique CDW formation process with dominated soft phonons and the static CDW occurring at different wavevectors, which are apparently different from that in the intensively kagome metals AV3Sb5 (A = K, Rb, Cs). This work not only unveiled very unique CDW mechanism, where competing CDW instabilities are strongly related to the wavevector-dependent electron-phonon coupling, but also clearly explained the electrical transport properties with the framework of electron-phonon coupling and extended phonon softening.

This is an in-time and rigorous work that could stimulate much interest not only in further understanding the CDW in ScV6Sn6, but also in other related intriguing physics, for example, the possible nematicity. It may warrant its publication in Nature communications after addressing the following concerns:

1. The $l=1/3$ CDW peaks in Fig. 2a look somewhat strange. For example, it is clear that the CDW peaks change very large between 88 K and 90 K. The authors explained this by involving “distribution of short-range qs-CDW clusters, domain formation due to the lowering of lattice symmetry below the CDW temperature, or a small sample misalignment”. Is there any concrete evidence for these arguments?
2. Why the data at 90 K in Fig. 2d is shown only from 0.4 to 0.6, whereas it in Fig. 2a shows over a much larger range?
3. A line along K-H should be included in Fig.4d (including the experimental data points), which could guide the easy capture of the key information in this figure. In addition, it is better to re-arrange the panels so Fig. 4d could be seen more clearly in details.
4. A very similar work appeared very recently on arXiv (arXiv:2304.09173) also focused on the competing CDW instabilities. A natural question is that, are the results reported herein consistent with that? In particular, was the softening of a flat phonon mode observed in IXS measurements?
5. The first-principles calculations indicate that q^* -CDW is energetically favorable as the ground state, while the experimentally observed one is the q -dependent. The reason is ascribed to the wavevector-dependent electron-phonon coupling. Is it possible to include the wavevector-dependent electron-phonon coupling into the first-principles calculations and then compare the energy again between the q^* -CDW and q -CDW?

Minor issues:

1. On page 2, what is the full spelling of the DHO model? And where the reference could be found?

2. Basic characterizations on the quality of the ScV6Sn6 crystals are necessary, for example, single crystal XRD or something else.

Reviewer #2 (Remarks to the Author):

Cao et al., performed comprehensive study of a new charged ordered kagome metal ScV6Sn6. Symmetry breaking orders in kagome metals is a topical research direction in the hard condensed matter community. ScV6Sn6 is a newly discovered kagome metal that features charge density wave with a wavevector different from AV3Sb5 and FeGe families. While the absence of other symmetry-breaking orders, such as superconductivity and magnetism, potentially undermines its fundamental interests, ScV6Sn6 offered an opportunity to uncover the complex landscape of CDW mechanisms in various kagome metals.

In this work, the authors combined inelastic x-ray scattering, angle-resolved photoemission spectroscopy and first principles calculations to understand the origin of CDW in ScV6Sn6. Their main experimental observation is that the large phonon softening at $(1/3, 1/3, 1/2)$ is different from the CDW wave-vector at $(1/3, 1/3, 1/3)$. Based on first principles calculations, the authors concluded that while the phonon self-energy effects is largest at $(1/3, 1/3, 1/2)$, the electron phonon coupling is much stronger at $(1/3, 1/3, 1/3)$ yielding a different CDW wavevector.

Overall, I think it is a nice work. The experimental observations are interesting. My main concern of the current manuscript is the calculations of phonon self-energy and q-dependent electron phonon coupling. The calculated λ_q appears unphysical large and sharp assuming the charge susceptibility is smooth. I speculate that this strange behavior arises from the incorrect application of the second equation in page 5, where λ_q is proportional to $1/\omega_q$. As shown in Fig. 4e, the DFT calculated phonon energy is zero near q_s , yielding a divergent λ_q .

Response to the referees

We thank the referees for reviewing our manuscript and providing constructive comments and critiques. We respond to all of the referees' comments/critiques in detail below, and have made corresponding changes in the revised manuscript (highlighted in red).

Response the first referee

“By measuring the inelastic X-ray scattering (IXS) and performing first-principles calculations, , The authors unveiled therein competing CDW instabilities in kagome metal ScV₆Sn₆, which leads to unique CDW formation process with dominated soft phonons and the static CDW occurring at different wavevectors, which are apparently different from that in the intensively kagome metals AV₃Sb₅ (A = K, Rb, Cs). This work not only unveiled very unique CDW mechanism, where competing CDW instabilities are strongly related to the wavevector-dependent electron-phonon coupling, but also clearly explained the electrical transport properties with the framework of electron-phonon coupling and extended phonon softening. This is an in-time and rigorous work that could stimulate much interest not only in further understanding the CDW in ScV₆Sn₆, but also in other related intriguing physics, for example, the possible nematicity. It may warrant its publication in Nature communications after addressing the following concerns:”

We thank the referee for reviewing our work, accurately summarizing our research, and finding that it may warrant publication in Nature Communications. We respond to the referee's concerns in detail below.

1. The $l=1/3$ CDW peaks in Fig. 2a look somewhat strange. For example, it is clear that the CDW peaks change very large between 88 K and 90 K. The authors explained this by involving “distribution of short-range q_s -CDW clusters, domain formation due to the lowering of lattice symmetry below the CDW temperature, or a small sample misalignment”. Is there any concrete evidence for these arguments?

We thank the referee for raising this interesting point. We agree with the referee that the $l=1/3$ q_s -CDW peaks appear strange. While our 88K and 90K data in Fig. 2a were both measured upon warming, they were not measured in the same warming process. Although we cannot fully rule out sample misalignments, such a scenario is unlikely as the main Bragg peaks are where they are expected. Therefore, the clear difference between the two scans suggest they probe distinct short-range clusters or domains of the q_s -CDW.

As the referee noted, there are reports of possible nematicity in this system, which provides a natural explanation for why ScV₆Sn₆ would exhibit distinct q_s -CDW domains/short-range clusters. In the presence of nematicity, the rotational symmetry is lowered, and due to coupling with the lattice, leads to distinct

domains. In terms of the scattering pattern for a single crystal sample, this results in the splitting of a single Bragg peak in the high-symmetry phase into multiple peaks in the nematic state (e. g. the nematic state of the iron-based superconductors [PRB 79, 180508 (2009)]), which then accounts for the difference between our 88K and 90K data: they probe different domains associated with split Bragg peaks.

2. Why the data at 90 K in Fig. 2d is shown only from 0.4 to 0.6, whereas it in Fig. 2a shows over a much larger range?

We thank the referee for raising this point. The 90K data in Fig. 2a and Fig. 2d were performed separately with different attenuators and over different scan ranges.

The 90 K data in Fig. 2a were measured over a wide scan range and using an aluminum attenuator with 1.8% transmission to prevent detector saturation caused by the intense scattering at $L=1/3$. However, using this attenuator leads to essentially no measurable signal at $L=1/2$. To accurately determine whether there is a peak at $L=1/2$ at 90K, we used an attenuator with 51.2% transmission and obtained the data in Fig. 2d over a small HH range of [0.4,0.6], in order to avoid detector saturation at $L=1/3$. The attenuation in both scans have been appropriately corrected for (via dividing the detector intensity by the attenuator transmission).

In the revised manuscript, the data in Fig. 2d are plotted using square symbols to highlight the different attenuator in this scan, compared to the scan in Fig. 2a. The captions have been updated correspondingly.

3. A line along K-H should be included in Fig.4d (including the experimental data points), which could guide the easy capture of the key information in this figure. In addition, it is better to re-arrange the panels so Fig. 4d could be seen more clearly in details.

We have modified Fig. 4, with the line along K-H added, following the referee's suggestion. The panels have also been re-arranged for improved readability.

4. A very similar work appeared very recently on arXiv (arXiv:2304.09173) also focused on the competing CDW instabilities. A natural question is that, are the results reported herein consistent with that? In particular, was the softening of a flat phonon mode observed in IXS measurements?

We thank the referee for raising this point. Our experimental data are fully consistent with those in arXiv:2304.09171. In both works, it is observed that the short-range fluctuating q^* -CDW forms at high temperatures, upon cooling the q^* -CDW is enhanced in intensity, and the associated phonons soften. As

the q_s -CDW emerges, the short-range CDW starts to diminish and eventually disappears at lower temperatures, revealing the competition between the two CDWs.

Quantitatively, the two works are also in good agreement. arXiv:2304.09173 finds the phonons at q^* fully soften to zero energy in following a mean-field behavior. Analyzing our data in a similar fashion (plotting $E = \sqrt{E_0^2 - \frac{\gamma^2}{4}}$ and fitting to a power-law behavior), we find that our data can be described by the same model used in arXiv:2304.09173, with similar values of the critical exponent and temperature.

Phonon softening at q_s were also detected in both works, with good agreement in terms of the phonon energy. For measurements at $q=(0,0,0.5)$, our data were limited to below ~ 110 K, and within this temperature range we did not find significant changes with temperatures. This is also consistent with the measurements in arXiv:2304.09173 at $Q=(0,0,6.5)$, where changes with temperature occur mostly above ~ 150 K.

Overall, our experimental data are in excellent agreement with those in arXiv:2304.09173 (see comparison in the figure below), although our measurements are more focused at the two vectors q_s and q^* , and did not directly probe whether there is the softening of a flat mode.

arXiv:2304.09173 data

Our IXS data

$\alpha = 0.47 \pm 0.04$ $T_c = 104 \pm 2$ K

5. The first-principles calculations indicate that q^* -CDW is energetically favorable as the ground state, while the experimentally observed one is the q -dependent. The reason is ascribed to the wavevector-dependent electron-phonon coupling. Is it possible to include the wavevector-dependent electron-phonon coupling into the first-principles calculations and then compare the energy again between the q^* -CDW and q -CDW?

We agree with the referee that a more realistic modeling of the CDW is possible if the q -dependent electron-phonon coupling could be fully taken into consideration. However, such a calculation requires the explicit consideration of electron-hole pairing, and such many-body effects are beyond the conventional DFT.

Despite the difficulty mentioned above, it is possible to roughly estimate the gain in energy when q -dependent electron-phonon coupling is considered, via evaluating the second-order correction of the Frölich Hamiltonian (first-order correction vanishes by definition):

$$\Delta E = \langle \Psi_0 | H_{elph} (E_0 - H_0)^{-1} H_{elph} | \Psi_0 \rangle,$$

where H_{elph} and H_0 are the electron-phonon coupling Hamiltonian and unperturbed Hamiltonian, respectively. Ψ_0 and E_0 are the ground state wavefunction and energy, respectively. The second order correction leads to

$$\Delta E \sim 2 \sum_{m,n,k} |g_{mn\nu}(k, q)|^2 \frac{(1-f_{nk})f_{mk+q}}{\epsilon_{mk+q} - \epsilon_{nk} - \omega_{q\nu}},$$

where 2 is due to the spin degeneracy. Using this formula, we estimate the energy gained by considering q -dependent electron-phonon coupling to be ~ 6.8 meV at q^* and ~ 36.9 meV at q_s for the lowest phonon modes, respectively. This then leads to the q_s -CDW being more competitive than the q^* -CDW, consistent with the ground state found experimentally. We note that the above analysis only provides a very rough estimate, and more detailed studies are needed to accurately model the ground state and the landscape of competing orders in ScV_6Sn_6 .

Minor issues:

1. On page 2, what is the full spelling of the DHO model? And where the reference could be found?

Following the referee's suggestion, we have added the full text and references for the first instance of DHO (damped harmonic oscillator) in the manuscript.

2. Basic characterizations on the quality of the ScV6Sn6 crystals are necessary, for example, single crystal XRD or something else.

We thank the referee for this suggestion. The left panel of the figure below shows an image of our ScV₆Sn₆ single crystal, which is a platelet with clear hexagonal cleaving edges in the ab-plane. We have also measured the Laue pattern, revealing a clear sixfold symmetric diffraction pattern expected for ScV₆Sn₆. These in combination with the electrical transport in Fig. 1(c) with relatively large RRR value, collectively indicate that our ScV₆Sn₆ single crystals phase pure and of high quality. This is further corroborated by the consistency between our IXS data and those in arXiv:2304.09173.

Response the second referee

Cao et al., performed comprehensive study of a new charged ordered kagome metal ScV6Sn6. Symmetry breaking orders in kagome metals is a topical research direction in the hard condensed matter community. ScV6Sn6 is a newly discovered kagome metal that features charge density wave with a wavevector different from AV3Sb5 and FeGe families. While the absence of other symmetry-breaking orders, such as superconductivity and magnetism, potentially undermines its fundamental interests, ScV6Sn6 offered an opportunity to uncover the complex landscape of CDW mechanisms in various kagome metals.

In this work, the authors combined inelastic x-ray scattering, angle-resolved photoemission spectroscopy and first principles calculations to understand the origin of CDW in ScV6Sn6. Their main experimental observation is that the large phonon softening at $(1/3, 1/3, 1/2)$ is different from the CDW wave-vector at $(1/3, 1/3, 1/3)$. Based on first principles calculations, the authors concluded that while the phonon self-energy effects is largest at $(1/3, 1/3, 1/2)$, the electron phonon coupling is much stronger at $(1/3, 1/3, 1/3)$ yielding a different CDW wavevector.

Overall, I think it is a nice work. The experimental observations are interesting. My main concern of the current manuscript is the calculations of phonon self-energy and q-dependent electron phonon coupling. The calculated λ_q appears unphysical large and sharp assuming the charge susceptibility is smooth. I speculate that this strange behavior arises from the incorrect application of the second equation

in page 5, where λ_q is proportional to $1/\omega_q$. As shown in Fig. 4e, the DFT calculated phonon energy is zero near q_s , yielding a divergent λ_q .

We thank the referee for reviewing our work, accurately summarizing our findings, and finding that our observations are interesting. We address the referee’s concern regarding our calculations below, and have made corresponding changes in our revised manuscript.

We thank the referee for raising the issue regarding the small phonon frequency at $(1/3, 1/3, 1/3)$. The obtained phonon frequency at $(1/3, 1/3, 1/3)$ in our calculations is 0.78 meV, which is not particularly small. To further show that the peak is not a singular behavior of ω_{qv} , we also plotted the momentum dependent $\lambda_{qv} |\omega_{qv}|$ for the lowest phonon mode. This quantity also peaks at $(1/3, 1/3, 1/3)$ (right figure below), and is free from the issue mentioned by the referee. Importantly, this quantity exhibits a *consistent and significant* enhancement at q-points close to q_s , so it is unlikely to be an artifact of the small phonon energy at $(1/3, 1/3, 1/3)$.

In addition, we also calculated the trace of electron-phonon coupling vertex $\sqrt{\sum |g|^2}$ for the lowest phonon mode at different q, which also peaks at $(1/3, 1/3, 1/3)$ (blue points in the left figure below, labelled as “|g|”). Therefore, we conclude the electron-phonon coupling strength indeed peaks at $q_s=(1/3, 1/3, 1/3)$, where the system is found to order experimentally.

We note that the calculated electron-phonon coupling strength is extremely large, because the calculation was performed on a structure without the CDW distortion. Since DFT calculations are conceptually performed at 0K, the huge EPC strength simply means the structure is unstable at low temperatures and a structural distortion would take place.

In the revised manuscript, we have added a plot of $\lambda_q |\omega_q|$ in Fig. 4(e), and added text to clarify the peak in λ_q at q_s is not due to the $1/\omega_q$ factor in the expression for λ_q . We have corrected errors in Fig. 4(e) and the text related to the dimension of λ_q (it is corrected to be dimensionless).

List of changes in the manuscript

1. Moved data in Supplementary Fig. 3(d) into Fig. 4(d) of the main text. Revised the presentation of Fig. 4, so that it is now larger and easier to read.
2. Updated the symbols in Fig. 2(d), so that the 106K data is plotted using circles, consistent with the 106K data Fig. 2(a); whereas the 90K data in Fig. 2(d) are plotted using squares to differentiate from the 90K data in Fig. 2(a).
3. Fig. 4(e) now also shows $\lambda_q |\omega_q|$, in addition to λ_q . The captions and text have been updated correspondingly. We have added a sentence in the main text stating that the peak in λ_q is not due the $1/\omega_q$ factor in the expression for λ_q . We have also corrected the units of λ_q to dimensionless.
4. Added the full description 'damped harmonic oscillator (DHO)' and references for the first instance of DHO.
5. Updated references that have been published in journals.
6. Added statements regarding error bars in the figure captions.

REVIEWER COMMENTS

Reviewer #1 (Remarks to the Author):

The authors well addressed all my concerns and clearly clarified the differences and similarities between their work and the one published on arXiv (2304.09173). The present version is satisfied. I would like to recommend the publication in NC.

Reviewer #2 (Remarks to the Author):

In the revised manuscript, the authors showed the electron-phonon coupling vertex, g , which still displays a divergent behavior. I found the calculated result still unphysical. As the authors replied, the divergence could be related to the unstable structure at zero temperature. If that is the case, I would expect that similar divergent behavior will be observed along L-A direction.

As a cross check, the author may also want to calculate g with high electron temperatures.

Response to the second referee

“In the revised manuscript, the authors showed the electron-phonon coupling vertex, g , which still displays a divergent behavior. I found the calculated result still unphysical. As the authors replied, the divergence could be related to the unstable structure at zero temperature. If that is the case, I would expect that similar divergent behavior will be observed along L-A direction.

As a cross check, the author may also want to calculate g with high electron temperatures.”

We thank the referee for raising these important points, and suggesting a viable route to check our results. We agree with the referee that the small frequency ω_q is a possible source for the divergent behavior in λ_q and $\lambda_q|\omega_q|$ (left of Fig. R2). Following the referee’s suggestions, we have carried out additional calculations which show that the q -dependent electron-phonon coupling (EPC) indeed peaks around q_s , which likely promotes the selection of q_s -CDW as the ground state.

Whereas our original calculations were performed at $T=0.01$ Ry, we performed additional calculations at an elevated electron temperature $T=0.1$ Ry (close to the electron temperature in arXiv:2305.15469), following the referee’s advice. As shown in Fig. R1 below, the structure is completely stable and the phonon frequency is nearly constant along K-H, for $T=0.1$ Ry. While the structure is stable without imaginary phonons, a broad hump in λ_q is still present around q_{CDW}/q_s . Moreover, a small hump in the phonon linewidth Π'' is observed close to q_{CDW}/q_s , which becomes more evident as T decreases from 0.1 Ry to 0.01 Ry. The phonon linewidth Π'' is the imaginary part of the phonon self-energy within the Migdal approximation, which is directly associated with the EPC. The enhancement of Π'' near q_s as temperature decreases is consistent with a q -dependent EPC that peaks around q_s promoting the q_s -CDW as the ground state.

In summary, there are two pieces of evidence that favor a q -dependent EPC peaking around q_s that promotes q_s -CDW: (1) the peak in λ_q persists to high electron temperatures, and (2) a hump in Π'' around q_s that becomes enhanced upon cooling .

Fig. R1

For comparison, we performed similar calculations along the high symmetry line A-L-H-A, where imaginary phonons appear at low temperatures (Fig. R2). For $T=0.01$ Ry (left of Fig. R2), we observe two peaks in λ_q and $\lambda_q|\omega_q|$ along A-L-H-A, where the phonon frequencies $\omega_q \rightarrow 0$. Despite these peaks, the phonon linewidth Γ'' exhibit local minima around these positions, and there are no peaks in λ_q when the electron temperature is increased to 0.1 Ry (right of Fig. R2).

The above comparison indicates that as the referee suggested, $\omega_q \rightarrow 0$ may lead to peaks in λ_q and $\lambda_q|\omega_q|$. In such a situation, calculations of Γ'' and at elevated electron temperatures act as tests for whether such peaks are indeed reflective of a q-dependent EPC. In the case of peaks on the A-L-H-A line, our findings indicate they result from $\omega_q \rightarrow 0$. In contrast, for q_s on the H-K path, our results [points (1) and (2) above] indicate a q-dependent EPC that is maximized around q_s , which contributes to selecting q_s -CDW as the ground state.

Fig. R2

Based on the points raised by the referee and our additional calculations, we believe the q-dependent EPC plays an important role in selecting q_s -CDW as the ground state (while softest phonons occur at q^*). We also acknowledge that the precise mechanism of q_s -CDW poses an interesting and potentially challenging problem that motivates more detailed theoretical studies.

We have revised our manuscript (highlighted in red) to make it clear why our findings suggest the presence of a q-dependent EPC, which promotes the selection of q_s -CDW as the ground state. Corresponding changes are also made in the figures and the Supplementary Information (added Supplementary Figs. 6-7 and Supplementary Note 4).

REVIEWERS' COMMENTS

Reviewer #2 (Remarks to the Author):

The new calculation at $T=0.t$ Ry on a stable phonon energy makes more sense to me. Intuitively, EPC coupling must be the key for the emergence of CDW phase in a non-magnetic system. But the proposed mechanism of this manuscript is still debatable. Nevertheless, given the interesting experimental observations, I would support its publications without further reviews.

I, however, would make a suggestion for the authors to consider. In my perspective, the non-physical divergence shown in Fig. 4(e) still very misleading and can potentially receive criticisms from the community. It might be a good idea to show the calculations at $T=0.1$ Ry.

Response to the second referee

“The new calculation at $T=0.t$ Ry on a stable phonon energy makes more sense to me. Intuitively, EPC coupling must be the key for the emergence of CDW phase in a non-magnetic system. But the proposed mechanism of this manuscript is still debatable. Nevertheless, given the interesting experimental observations, I would support its publications without further reviews.

I, however, would make a suggestion for the authors to consider. In my perspective, the non-physical divergence shown in Fig. 4(e) still very misleading and can potentially receive criticisms from the community. It might be a good idea to show the calculations at $T=0.1$ Ry.”

We thank the referee for recommending our manuscript for publication and providing further constructive suggestions. We agree that a complete understanding of the CDW-formation mechanism in ScV_6Sn_6 warrants further studies, although the q-dependent EPC undoubtedly plays an important role, as demonstrated in the present work.

In the revised manuscript, we have updated Fig. 4(e) to show calculations at $T=0.1\text{Ry}$, as suggested by the referee. The descriptions and discussions in the main text have been updated correspondingly.